# MA2QL: A Minimalist Approach to Fully Decentralized Multi-Agent Reinforcement Learning

## Abstract

Decentralized learning has shown great promise for cooperative multi-agent reinforcement learning (MARL). However, non-stationarity remains a significant challenge in fully decentralized learning. In the paper, we tackle the non-stationarity problem in the simplest and fundamental way and propose *multi-agent alternate Q-learning* (MA2QL), where agents take turns to update their Q-functions by Q-learning. MA2QL is a *minimalist* approach to fully decentralized cooperative MARL but is theoretically grounded. We prove that when each agent guarantees $\varepsilon$-convergence at each turn, their joint policy converges to a Nash equilibrium. In practice, MA2QL only requires minimal changes to independent Q-learning (IQL). We empirically evaluate MA2QL on a variety of cooperative multi-agent tasks. Results show MA2QL consistently outperforms IQL, which verifies the effectiveness of MA2QL, despite such minimal changes.

## 1 Introduction

Cooperative multi-agent reinforcement learning (MARL) is a well-abstracted model for a broad range of real applications, including logistics (Li et al., 2019), traffic signal control (Xu et al., 2021), power dispatch (Wang et al., 2021b), and inventory management (Feng et al., 2022). In cooperative MARL, centralized training with decentralized execution (CTDE) is a popular learning paradigm, where the information of all agents can be gathered and used in training. Many CTDE methods (Lowe et al., 2017; Foerster et al., 2018; Sunehag et al., 2018; Rashid et al., 2018; Wang et al., 2021a; Zhang et al., 2021; Su & Lu, 2022; Li et al., 2022) have been proposed and shown great potential to solve cooperative multi-agent tasks.

Another paradigm is decentralized learning, where each agent learns its policy based on only local information. Decentralized learning is less investigated but desirable in many scenarios where the information of other agents is not available, and for better robustness, scalability, and security (Zhang et al., 2019). However, *fully decentralized learning* of agent policies (*i.e.*, without communication) is still an open challenge in cooperative MARL.

The most straightforward way for fully decentralized learning is directly applying independent learning at each agent (Tan, 1993), which however induces the well-known non-stationarity problem for all agents (Zhang et al., 2019) and may lead to learning instability and a non-convergent joint policy, though the performance varies as shown in empirical studies (Rashid et al., 2018; de Witt et al., 2020; Papoudakis et al., 2021; Yu et al., 2021).

In the paper, we directly tackle the non-stationarity problem in the simplest and fundamental way, *i.e.*, fixing the policies of other agents while one agent is learning. Following this principle, we propose **multi-agent alternate Q-learning (MA2QL)**, a *minimalist* approach to fully decentralized cooperative multi-agent reinforcement learning, where agents take turns to update their policies by Q-learning. MA2QL is theoretically grounded and we prove that when each agent guarantees $\varepsilon$-convergence at each turn, their joint policy converges to a Nash equilibrium. In practice, MA2QL only requires minimal changes to independent Q-learning (IQL) (Tan, 1993; Tampuu et al., 2015) and also independent DDPG (Lillicrap et al., 2016) for continuous action, *i.e.*, simply swapping the order of two lines of codes as follows. Their major difference can be highlighted as: *MA2QL agents take turns to update Q-functions by Q-learning, whereas IQL agents simultaneously update Q-functions by Q-learning.*

**IQL**

```
1: repeat
2:     all agents interact in the environment
3:     for i ← 1, n do
4:         agent i updates by Q-learning
5:     end for
6: until terminate
```

**MA2QL**

```
1: repeat
2:     for i ← 1, n do
3:         all agents interact in the environment
4:         agent i updates by Q-learning
5:     end for
6: until terminate
```

We evaluate MA2QL on a didactic game to empirically verify its convergence, and multi-agent particle environments (Lowe et al., 2017), multi-agent MuJoCo (Peng et al., 2021), and StarCraft multi-agent challenge (Samvelyan et al., 2019) to verify its performance with discrete and continuous action spaces, and fully and partially observable environments. We find that MA2QL consistently outperforms IQL, despite such minimal changes. The effectiveness of MA2QL suggests that simpler approaches may have been left underexplored for fully decentralized cooperative multi-agent reinforcement learning.

## 2 BACKGROUND

### 2.1 PRELIMINARIES

**Dec-POMDP.** Decentralized partially observable Markov decision process (Dec-POMDP) is a general model for cooperative MARL. A Dec-POMDP is a tuple $M = \{S, A, P, Y, O, I, n, r, \gamma\}$. $S$ is the state space, $n$ is the number of agents, $\gamma \in [0, 1)$ is the discount factor, and $I = \{1, 2 \cdots n\}$ is the set of all agents. $A = A_1 \times A_2 \times \cdots \times A_n$ represents the joint action space where $A_i$ is the individual action space for agent $i$. $P(s'|s, \boldsymbol{a}) : S \times A \times S \to [0, 1]$ is the transition function, and $r(s, \boldsymbol{a}) : S \times A \to \mathbb{R}$ is the reward function of state $s$ and joint action $\boldsymbol{a}$. $Y$ is the observation space, and $O(s, i) : S \times I \to Y$ is a mapping from state to observation for each agent. The objective of Dec-POMDP is to maximize $J(\boldsymbol{\pi}) = \mathbb{E}_{\boldsymbol{\pi}} \left[ \sum_{t=0}^{\infty} \gamma^t r(s_t, \boldsymbol{a}_t) \right]$, and thus we need to find the optimal joint policy $\boldsymbol{\pi}^* = \arg\max_{\boldsymbol{\pi}} J(\boldsymbol{\pi})$. To settle the partial observable problem, history $\tau_i \in \mathcal{T}_i : (Y \times A_i)^*$ is often used to replace observation $o_i \in Y$. Each agent $i$ has an individual policy $\pi_i(a_i|\tau_i)$ and the joint policy $\boldsymbol{\pi}$ is the product of each $\pi_i$. Though the individual policy is learned as $\pi_i(a_i|\tau_i)$ in practice, as Dec-POMDP is undecidable (Madani et al., 1999) and the analysis in partially observable environments is much harder, we will use $\pi_i(a_i|s)$ in analysis and proofs for simplicity.

**Dec-MARL.** Although decentralized cooperative multi-agent reinforcement learning (Dec-MARL) has been previously investigated (Zhang et al., 2018; de Witt et al., 2020), the setting varies across these studies. In this paper, we consider Dec-MARL as a *fully* decentralized solution to Dec-POMDP, where each agent learns its policy/Q-function from its own action individually ***without communication or parameter-sharing***. Therefore, in Dec-MARL, each agent $i$ actually learns in the environment with transition function $P_i(s'|s, a_i) = \mathbb{E}_{a_{-i} \sim \pi_{-i}}[P(s'|s, a_i, a_{-i})]$ and reward function $r_i(s, a_i) = \mathbb{E}_{a_{-i} \sim \pi_{-i}}[r(s, a_i, a_{-i})]$, where $\pi_{-i}$ and $a_{-i}$ respectively denote the joint policy and joint action of all agents expect $i$. As other agents are also learning (*i.e.*, $\pi_{-i}$ is changing), from the perspective of each individual agent, the environment is non-stationary. This is the non-stationarity problem, the main challenge in Dec-MARL.

**IQL.** Independent Q-learning (IQL) is a straightforward method for Dec-MARL, where each agent $i$ learns a Q-function $Q(s, a_i)$ by Q-learning. However, as all agents learn simultaneously, there is no theoretical guarantee on convergence due to non-stationarity, to the best of our knowledge. In practice, IQL is often taken as a simple baseline in favor of more elaborate MARL approaches, such as value-based CTDE methods (Rashid et al., 2018; Son et al., 2019). However, much less attention has been paid to IQL itself for Dec-MARL.

### 2.2 MULTI-AGENT ALTERNATE POLICY ITERATION

To address the non-stationarity problem in Dec-MARL, a fundamental way is simply to make the environment stationary during the learning of each agent. Following this principle, we let agents learn by turns; in each turn, one agent performs policy iteration while fixing the policies of other agents. This procedure is referred to as *multi-agent alternate policy iteration*. As illustrated in Figure 1,

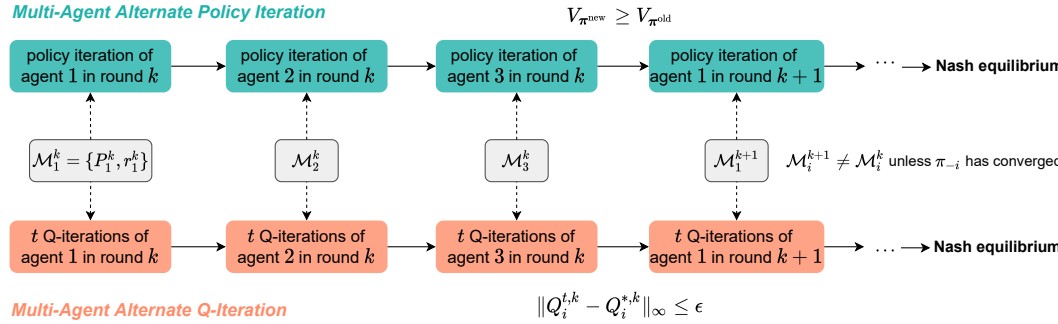

Figure 1: Illustration of *multi-agent alternate policy iteration* (upper panel) and *multi-agent alternate Q-iteration* (lower panel) of three agents. As essentially the MDP differs at different turns of each agent, policy iteration/Q-iteration of each agent iterates over different MDPs.

multi-agent alternate policy iteration differs from policy iteration in single-agent RL. In single-agent RL, policy iteration is performed on the same MDP. However, here, for each agent, policy iteration at a different round is performed on a different MDP. As $\pi_{-i}$ is fixed at each turn, $P_i(s'|s, a_i)$ and $r_i(s, a_i)$ are stationary and we can easily have the following lemma.

**Lemma 1** (multi-agent alternate policy iteration). *If all agents take turns to perform policy iteration, their joint policy sequence $\{\boldsymbol{\pi}\}$ monotonically improves and converges to a Nash equilibrium.*

*Proof.* In each turn, as the policies of other agents are fixed, the agent $i$ has the following update rule for policy evaluation,

$$Q_{\pi_i}(s, a_i) \leftarrow r_i(s, a_i) + \gamma \mathbb{E}_{s' \sim P_i, a'_i \sim \pi_i}[Q_{\pi_i}(s', a'_i)]. \tag{1}$$

We can have the convergence of policy evaluation in each turn by the standard results (Sutton & Barto, 2018). Moreover, as $\pi_{-i}$ is fixed, it is straightforward to have

$$Q_{\pi_i}(s, a_i) = \mathbb{E}_{a_{-i} \sim \pi_{-i}}[Q_{\boldsymbol{\pi}}(s, a, a_i)]. \tag{2}$$

Then, the agent $i$ performs policy improvement by

$$\pi_i^{\text{new}}(s) = \arg \max_{a_i} \mathbb{E}_{\pi_{-i}^{\text{old}}}[Q_{\boldsymbol{\pi}^{\text{old}}}(s, a_i, a_{-i})]. \tag{3}$$

As the policies of other agents are fixed (i.e., $\pi_{-i}^{\text{new}} = \pi_{-i}^{\text{old}}$), we have

$$
\begin{aligned}
V_{\boldsymbol{\pi}^{\text{old}}}(s) &= \mathbb{E}_{\boldsymbol{\pi}^{\text{old}}}[Q_{\boldsymbol{\pi}^{\text{old}}}(s, a_i, a_{-i})] = \mathbb{E}_{\pi_i^{\text{old}}} \mathbb{E}_{\pi_{-i}^{\text{old}}}[Q_{\boldsymbol{\pi}^{\text{old}}}(s, a_i, a_{-i})] \\
&\leq \mathbb{E}_{\pi_i^{\text{new}}} \mathbb{E}_{\pi_{-i}^{\text{old}}}[Q_{\boldsymbol{\pi}^{\text{old}}}(s, a_i, a_{-i})] = \mathbb{E}_{\pi_i^{\text{new}}} \mathbb{E}_{\pi_{-i}^{\text{new}}}[Q_{\boldsymbol{\pi}^{\text{old}}}(s, a_i, a_{-i})] \\
&= \mathbb{E}_{\boldsymbol{\pi}^{\text{new}}}[Q_{\boldsymbol{\pi}^{\text{old}}}(s, a_i, a_{-i})] = \mathbb{E}_{\boldsymbol{\pi}^{\text{new}}}[r(s, a_i, a_{-i}) + \gamma V_{\boldsymbol{\pi}^{\text{old}}}(s')] \\
&\leq \cdots \leq V_{\boldsymbol{\pi}^{\text{new}}}(s),
\end{aligned}
\tag{4}
$$

where the first inequality is from (3). This proves that the policy improvement of agent $i$ in each turn also improves the joint policy. Thus, as agents perform policy iteration by turn, the joint policy sequence $\{\boldsymbol{\pi}\}$ improves monotonically, and $\{\boldsymbol{\pi}\}$ will converge to a Nash equilibrium since no agents can improve the joint policy unilaterally at convergence. □

Lemma 1 immediately indicates an approach for Dec-MARL with convergence guarantee and also tells us that if we find the optimal policy for agent $i$ in each round $k$ given the other agents' policies $\pi_{-i}^k$, then the joint policy will obtain the largest improvement. This result can be formulated as follows,

$$
\begin{aligned}
\pi_i^{*,k} &= \arg \max_{\pi_i} \mathbb{E}_{\pi_{-i}^k}\left[Q_{\pi_i, \pi_{-i}^k}(s, a_i, a_{-i})\right] \\
V_{\pi_i, \pi_{-i}^k}(s) &\leq V_{\pi_i^{*,k}, \pi_{-i}^k}(s) \quad \forall \pi_i, \forall s.
\end{aligned}
\tag{5}
$$

We could obtain this $\pi_i^{*,k}$ by policy iteration with many *on-policy* iterations. However, such a method will face the issue of sample inefficiency which may be amplified in MARL settings. We will use Q-iteration to settle this problem in the next section.

## 3 METHOD

To address the problem of multi-agent alternate policy iteration, we propose *multi-agent alternate Q-iteration*, which is sufficiently truncated for fast learning but still has the same theoretical guarantee. Further, based on multi-agent alternate Q-iteration, we derive *multi-agent alternate Q-learning*, which makes the minimal change to IQL to form a simple yet effective value-based decentralized learning method for cooperative MARL.

### 3.1 MULTI-AGENT ALTERNATE Q-ITERATION

Instead of policy iteration, we let agents perform Q-iteration by turns as depicted in Figure 1. Let $\mathcal{M}_i^k = \{P_i^k, r_i^k\}$ denote the MDP of agent $i$ in round $k$, where we have $\mathcal{M}_i^k \neq \mathcal{M}_i^{k-1}$ unless $\pi_{-i}$ has converged, and $Q_i^{t,k}(s, a_i)$ denote the Q-function of agent $i$ with $t$ updates in the round $k$. We define the Q-iteration as follows,

$$Q_i^{t+1,k}(s, a_i) \leftarrow r_i^k(s, a_i) + \gamma \mathbb{E}_{s' \sim P_i^k} \left[ \max_{a_i'} Q_i^{t,k}(s', a_i') \right]. \tag{6}$$

Then, the sequence $\{Q_i^{t,k}\}$ converges to $Q_i^{*,k}$ with respect to the MDP $\mathcal{M}_i^k = \{P_i^k, r_i^k\}$, and we have the following lemma and corollary.

**Lemma 2** ($\varepsilon$-convergent Q-iteration)**.** *By iteratively applying Q-iteration* (6) *at each agent $i$ for each turn, for any $\varepsilon > 0$, we have*

$$\left\| Q_i^{t,k} - Q_i^{*,k} \right\|_\infty \leq \varepsilon, \quad when \ \ t \geq \frac{\log\left((1-\gamma)\varepsilon\right) - \log(2R + 2\varepsilon)}{\log \gamma}, \tag{7}$$

*where $R = \frac{r_{\max}}{1-\gamma}$ and $r_{\max} = \max_{s,\boldsymbol{a}} r(s, \boldsymbol{a})$.*

**Corollary 1.** *For any $\varepsilon > 0$, if we take sufficient Q-iteration $t_i^k$, i.e., $Q_i^k = Q_i^{t_i^k, k}$, then we have*

$$\left\| Q_i^k - Q_i^{*,k} \right\|_\infty \leq \varepsilon \quad \forall k, i.$$

With Lemma 1, Lemma 2, and Corollary 1, we have the following theorem.

**Theorem 1** (multi-agent alternate Q-iteration)**.** *Suppose that $Q_i^*(s, \cdot)$ has the unique maximum for all states and all agents. If all agents in turn take Q-iteration to $\|Q_i^k - Q_i^{*,k}\|_\infty \leq \varepsilon$, then their joint policy sequence $\{\boldsymbol{\pi}^k\}$ converges to a Nash equilibrium, where $\pi_i^k(s) = \arg\max_{a_i} Q_i^k(s, a_i)$.*

All the proofs are included in Appendix A.

Theorem 1 assumes that for each agent, $Q_i^*$ has the unique maximum for all states. Although this may not hold in general, in practice we can easily settle this by introducing a positive random noise to the reward function. Suppose the random noise is bounded by $\delta$, then we can easily derive that the performance drop of optimizing environmental reward plus noise is bounded by $\delta/(1-\gamma)$. As we can make $\delta$ arbitrarily small, the bound is tight.

### 3.2 MULTI-AGENT ALTERNATE Q-LEARNING

From Theorem 1, we know that if each agent $i$ guarantees $\varepsilon$-convergence to $Q_i^{*,k}$ in each round $k$, multi-agent alternate Q-iteration also guarantees a Nash equilibrium of the joint policy. This immediately suggests a simple, practical fully decentralized learning method, namely multi-agent alternate Q-learning (MA2QL).

For learning Q-table or Q-network, MA2QL makes minimal changes to IQL.

- For learning Q-tables, all agents in turn update their Q-tables. At a round $k$ of an agent $i$, all agents interact in the environment, and the agent $i$ updates its Q-table a few times using the collected transitions $\langle s, a_i, r, s' \rangle$.
- For learning Q-networks, all agents in turn update their Q-networks. At a round of an agent $i$, all agents interact in the environment, and each agent $j$ stores the collected transitions $\langle s, a_j, r, s' \rangle$ into its replay buffer, and the agent $i$ updates its Q-network using sampled mini-batches from its replay buffer.

There is a slight difference between learning Q-table and Q-network. Strictly following multi-agent alternate Q-iteration, Q-table is updated by transitions sampled from the current MDP. On the other hand, Q-network is updated by mini-batches sampled from the replay buffer. If the replay buffer only contains the experiences sampled from the current MDP, learning Q-network also strictly follows multi-agent alternate Q-iteration. However, in practice, we slightly deviate from that and allow the replay buffer to contain transitions of past MDPs, following IQL (Sunehag et al., 2018; Rashid et al., 2018; Papoudakis et al., 2021) for sample efficiency, the convergence may not be theoretically guaranteed though.

MA2QL and IQL can be simply summarized and highlighted as ***MA2QL agents take turns to update Q-functions by Q-learning, whereas IQL agents simultaneously update Q-functions by Q-learning***.

## 4 RELATED WORK

**CTDE.** The most popular learning paradigm in cooperative MARL is centralized training with decentralized execution (CTDE), including value decomposition and multi-agent actor-critic. For value decomposition (Sunehag et al., 2018; Rashid et al., 2018; Son et al., 2019; Wang et al., 2021a), a joint Q-function is learned in a centralized manner and factorized into local Q-functions to enable decentralized execution. For multi-agent actor-critic, a centralized critic, Q-function or V-function, is learned to provide gradients for local policies (Lowe et al., 2017; Foerster et al., 2018; Yu et al., 2021). Moreover, some studies (Wang et al., 2020; Peng et al., 2021; Su & Lu, 2022) combine value decomposition and multi-agent actor-critic to take advantage of both, while others rely on maximum-entropy RL to naturally bridge the joint Q-function and local policies (Iqbal & Sha, 2019; Zhang et al., 2021).

**Decentralized learning.** Another learning paradigm in cooperative MARL is decentralized learning, where the simplest way is for each agent to learn independently, *e.g.*, independent Q-learning (IQL) or independent actor-critic (IAC). These methods are usually taken as simple baselines for CTDE methods. For example, IQL is taken as a baseline in value decomposition methods (Sunehag et al., 2018; Rashid et al., 2018), while IAC is taken as a baseline in multi-agent actor-critic (Foerster et al., 2018; Yu et al., 2021). Some study further considers decentralized learning with communication for parameter-sharing (Zhang et al., 2018; Li et al., 2020). However, parameter-sharing should be considered as centralized training (Terry et al., 2020). More recently, IAC (*i.e.*, independent PPO) has been empirically investigated and found remarkably effective in several cooperative MARL tasks (de Witt et al., 2020; Yu et al., 2021), including multi-agent particle environments (MPE) (Lowe et al., 2017) and StarCraft multi-agent challenge (SMAC). However, as actor-critic methods follow the principle different from Q-learning, we will not focus on IAC for comparison in the experiment. On the other hand, IQL has also been thoroughly benchmarked and its performance is close to CTDE methods in a few tasks (Papoudakis et al., 2021). This sheds some light on the potential of value-based decentralized cooperative MARL. Although there are some Q-learning variants, *i.e.*, hysteretic Q-learning (Matignon et al., 2007) and lenient Q-learning (Palmer et al., 2018), for fully decentralized learning, they both are heuristic and their empirical performance is even worse than IQL (Zhang et al., 2020).

Unlike existing work, MA2QL is theoretically grounded for fully decentralized learning and in practice makes minimal changes to IQL. In the next section, we provide the empirical comparison between MA2QL and IQL.

## 5 EXPERIMENTS

In this section, we empirically study MA2QL on a set of cooperative multi-agent tasks, including a didactic game, multi-agent particle environments (MPE) (Lowe et al., 2017), multi-agent MuJoCo (Peng et al., 2021), and StarCraft multi-agent challenge (SMAC) (Samvelyan et al., 2019), to investigate the following questions.

**1.** *Does MA2QL converge and what does it converge to empirically, compared with the optimal solution and IQL? Does the number of Q-function updates affect the convergence?*

**2.** *As MA2QL only makes the minimal changes to IQL, is MA2QL indeed better than IQL in both discrete and continuous action spaces, and in more complex tasks?*

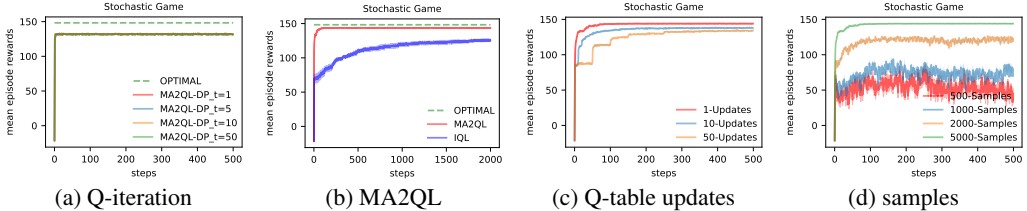

Figure 2: Empirical studies of MA2QL on the didactic game: (a) different numbers of Q-iterations performed by dynamic programming at each turn; (b) learning curve of MA2QL compared with IQL and the global optimum; (c) different numbers of Q-table updates at each turn; (d) different number of sampled transitions at each turn, where x-axis is learning steps.

In all the experiments, the training of MA2QL and IQL is based on the same number of environmental steps (*i.e.*, the same number of samples). Moreover, as the essential difference between MA2QL and IQL is that MA2QL agents take turns to update Q-function while IQL agents update Q-function simultaneously, for a fair comparison, the total number of Q-function updates for each agent in MA2QL is set to be the same with that in IQL. For example, in a setting of $n$ agents, if IQL agents update Q-function one step (*e.g.*, one gradient step) every environmental step, MA2QL agents update Q-function $n$ steps every $n$ environmental steps. For learning Q-networks, the size of the replay buffer is also the same for IQL and MA2QL. Again, we *do not use parameter-sharing*, which should not be allowed in decentralized settings (Terry et al., 2020). More detailed experimental settings, hyperparameters, and additional results are available in Appendix B, C, and D, respectively. All results are presented using the mean and standard deviation of five random seeds.

## 5.1 A DIDACTIC GAME

The didactic game is a cooperative stochastic game, which is randomly generated for the reward function and transition probabilities with 30 states, 3 agents, and 5 actions for each agent. Each episode in the game contains 30 timesteps. For comparison, we use dynamic programming to find the global optimal solution, denoted as OPTIMAL. For MA2QL and IQL, each agent independently learns a $30 \times 5$ Q-table.

First, we investigate how the number of Q-iterations empirically affects the convergence of multi-agent alternate Q-iteration, where Q-iteration is performed by dynamic programming and denoted as MA2QL-DP. As shown in Figure 2(a), we can see that different numbers of Q-iterations (*i.e.*, $t = 1, 5, 10, 50$) that each agent takes at each turn do not affect the convergence in the didactic game, even when $t = 1$. This indicates $\varepsilon$-convergence of Q-iteration can be easily satisfied with as few as one iteration.

Next, we compare the performance of MA2QL and IQL. As illustrated in Figure 2(b), IQL converges slowly (about 2000 steps), while MA2QL converges much faster (less than 100 steps) to a better return and also approximates OPTIMAL. Once again, MA2QL and IQL use the same number of samples and Q-table updates for each agent. One may notice that the performance of MA2QL is better than MA2QL-DP. This may be attributed to sampling and exploration of MA2QL, which induces a better Nash equilibrium.

Then, we investigate MA2QL in terms of the number of Q-table updates at each turn, which resembles the number of Q-iterations by learning on samples. Specifically, denoting $K$ as the number of Q-table updates, to update an agent, we repeat $K$ times of the process of sampling experiences and updating Q-table. This means with a larger $K$, agents take turns less frequently. As shown in Figure 2(c), with larger $K$, the learning curve is more stair-like, which means in this game a small number $K$ is enough for convergence at each turn. Thus, with larger $K$, the learning curve converges more slowly.

Last, we investigate how the number of collected transitions at each turn impacts the performance of MA2QL. As depicted in Figure 2(d), the performance of MA2QL is better with more samples. This is because, with more samples, the update of Q-learning using these samples is more like to induce a full iteration of Q-table.

In summary, as illustrated in Figure 2, the number of Q-function updates or samples may influence the performance of MA2QL. However, in the following experiments, we tune the performance of

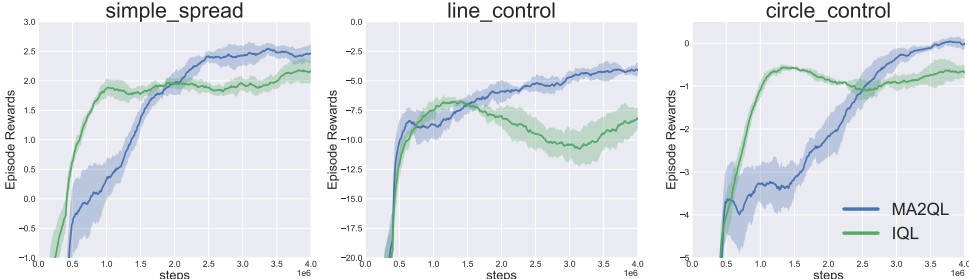

Figure 3: Learning curve of MA2QL compared with IQL in 5-agent simple spread, 5-agent line control, and 7-agent circle control in MPE, where x-axis is environment steps.

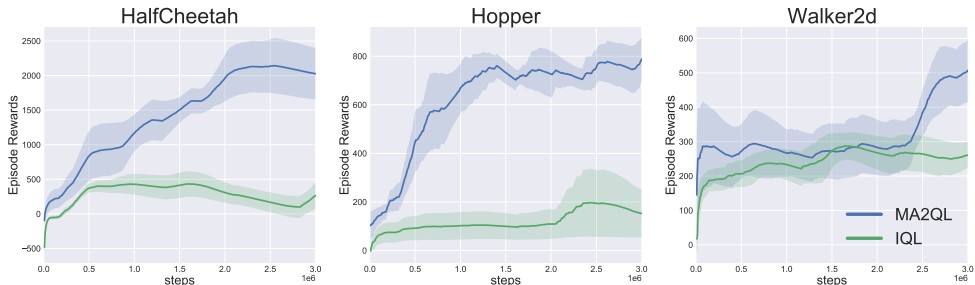

Figure 4: Learning curve of MA2QL compared with IQL in 2-agent HalfCheetah, 3-agent Hopper, and 3-agent Walker2d in multi-agent MuJoCo, where x-axis is environment steps.

IQL as best as we can and correspondingly change the configuration of MA2QL as stated above for a fair comparison.

## 5.2  MPE

MPE is a popular environment in cooperative MARL. We consider three partially observable tasks: 5-agent simple spread, 5-agent line control, and 7-agent circle control (Agarwal et al., 2020), where the action space is set to *discrete*. Moreover, we use the sparse reward setting for these tasks, thus they are more difficult than the original ones. More details are available in Appendix B. For both IQL and MA2QL, Q-network is learned by DQN (Mnih et al., 2013).

Figure 3 shows the learning curve of MA2QL compared with IQL in these three MPE tasks. In simple spread and circle control, at the early training stage, IQL learns faster and better than MA2QL, but eventually MA2QL converges to a better joint policy than IQL. IQL always converges to a worse sub-optimum than MA2QL, similar to that in the didactic game. Moreover, unlike the didactic game, simultaneous learning of IQL may also make the learning unstable even at the late training stage as in line control and circle control, where the episode rewards may decrease. On the other hand, learning by turns gradually improves the performance and converges to a better joint policy than IQL.

As MA2QL and IQL both use replay buffer that contains old experiences, ***why does MA2QL outperform IQL***? The reason is that their experiences are generated in different manners. In the Q-learning procedure for each agent $i$, the ideal target is $y_i = r_i^{\boldsymbol{\pi}}(s, a_i) + \mathbb{E}_{s' \sim P_i^{\boldsymbol{\pi}}(\cdot|s,a_i)}[\max_{a_i'} Q_i(s', a_i')]$ and the practical target is $\tilde{y}_i = r_i^{\boldsymbol{\pi}_{\mathrm{D}}}(s, a_i) + \mathbb{E}_{s' \sim P_i^{\boldsymbol{\pi}_{\mathrm{D}}}(\cdot|s,a_i)}[\max_{a_i'} Q_i(s', a_i')]$, where $\boldsymbol{\pi}_{\mathrm{D}}$ is the average joint policy for the experiences in the replay buffer. We then can easily obtain a bound for the target that $|y_i - \tilde{y}_i| \leq \frac{2-\gamma}{1-\gamma} r_{\max} D_{\mathrm{TV}} \left( \pi^{-i}(\cdot|s) \| \pi_{\mathrm{D}}^{-i}(\cdot|s) \right)$ where $r_{\max} = \max_{s,\boldsymbol{a}} r(s, \boldsymbol{a})$. We can then give an analysis from the aspect of the divergence between $\boldsymbol{\pi}$ and $\boldsymbol{\pi}_{\mathrm{D}}$. MA2QL obtains experiences with only one agent learning, so the variation for the joint policy is smaller than that of IQL. Thus, in general, the divergence between $\boldsymbol{\pi}$ and $\boldsymbol{\pi}_{\mathrm{D}}$ is smaller for MA2QL, which is beneficial to the learning.

## 5.3  MULTI-AGENT MUJOCO

Multi-agent MuJoCo has become a popular environment in cooperative MARL for *continuous* action space. We choose three robotic control tasks: 2-agent HalfCheetah, 3-agent Hopper, and 3-agent

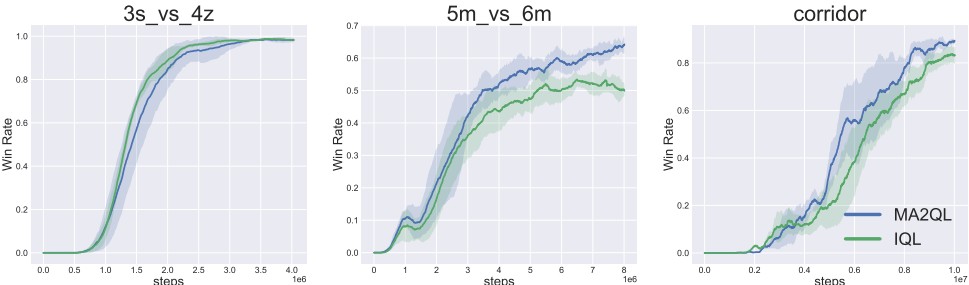

Figure 5: Learning curve of MA2QL compared with IQL on `3s_vs_4z` (easy), `5m_vs_6m` (hard) and `corridor` (super hard) in SMAC, where x-axis is environment steps.

Walker2d. To investigate continuous action space in both partially and fully observable environments, we configure 2-agent HalfCheetah and 3-agent Hopper as fully observable, and 3-agent Walker2d as partially observable. For both IQL and MA2QL, we use DDPG (Lillicrap et al., 2016) as the alternative to DQN to learn a Q-network and a deterministic policy for each agent to handle continuous action space.

In comparison to discrete action space, training multiple cooperative agents in continuous action space still remains challenging due to the difficulty of exploration and coordination in continuous action space. Thus, the evaluation on these multi-agent MuJoCo tasks can better demonstrate the effectiveness of decentralized cooperative MARL methods. As illustrated in Figure 4, in all the tasks, we find that MA2QL consistently and significantly outperforms IQL while IQL struggles. We believe the reason is that the robotic control tasks are much more dynamic than MPE and the non-stationarity induced by simultaneous learning of IQL may be amplified, which makes it hard for agents to learn effective and cooperative policies. On the other hand, alternate learning of MA2QL can deal with the non-stationarity and sufficiently stabilize the environment during the learning process, especially in HalfCheetah and Hopper, where MA2QL stably converges to much better performance than IQL. According to these experiments, we can verify the superiority of MA2QL over IQL in the continuous action space.

## 5.4 SMAC

SMAC is a popular partially observable environment for benchmarking cooperative MARL algorithms. SMAC has a much larger exploration space, where agents are much easy to get stuck in sub-optimal policies especially in the decentralized setting. We test our method on three representative maps for three difficulties: `3s_vs_4z` (easy), `5m_vs_6m` (hard), and `corridor` (super hard), where harder map has more agents. It is worth noting that we do not use any global state in the decentralized training and each agent learns on its own trajectory.

The results are shown in Figure 5. On the map `3s_vs_4z`, IQL and MA2QL both converge to the winning rate of 100%. However, on the hard and super hard map `5m_vs_6m` and `corridor`, MA2QL achieves stronger than IQL. Moreover, in these maps there are different numbers of agents, 3 agents in `3s_vs_4z`, 5 agents in `5m_vs_6m`, and 6 agents in `corridor`, thus the results may also indicate the good scalability of MA2QL, though the difficulty of the map varies.

It is worth noting that the recent study (Papoudakis et al., 2021) shows that IQL performs well in SMAC, even close to CTDE methods like QMIX (Rashid et al., 2018). Here, we show that MA2QL can still outperform IQL in three maps with various difficulties, which indicates that MA2QL can also tackle the non-stationarity problem and bring performance gain in more complex tasks.

## 5.5 STUDY ON HYPERPARAMETERS AND SCALABILITY

We further investigate the influence of the hyperparameters on MA2QL and the scalability of MA2QL. First, we study the effect of $K$ (the number of Q-network updates at each turn) in the robotic control task: 3-agent Hopper. We consider $K = [4000, 8000, 40000]$. As shown in Figure 6(a), when $K$ is small, it outperforms IQL but still gets stuck in sub-optimal policies. On the contrary, if $K$ is large, different $K$ affects the efficiency of the learning, but not the final performance.

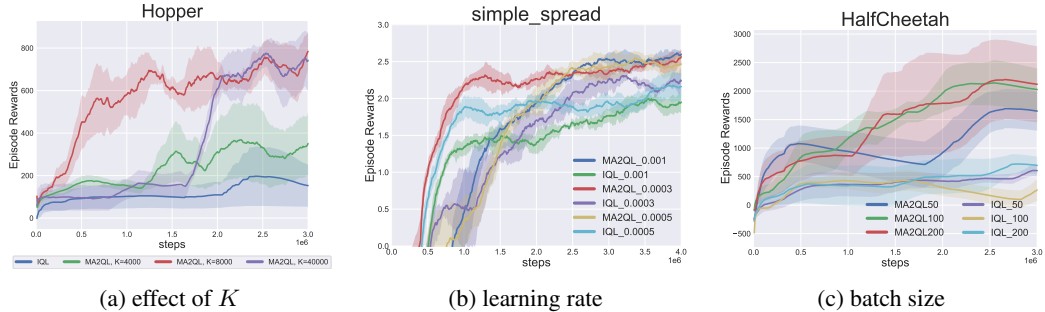

(a) effect of $K$         (b) learning rate         (c) batch size

Figure 6: The effect of hyperparameters: (a) learning curves of MA2QL with different $K$ in 3-agent Hopper, compared with IQL; (b) learning curve of MA2QL compared with IQL with different learning rates in simple spread in MPE; (c) learning curve of MA2QL compared with IQL with different batch sizes in HalfCheetah in multi-agent MuJoCo.

As MA2QL is essentially an add-on for IQL, the hyperparameters are the same for IQL and MA2QL. In the experiments, the hyperparameters of IQL are taken from previous studies (Achiam, 2018; Hu et al., 2021) for well-tuned performance, and we do not tune the hyperparameters of IQL for MA2QL. To further study the effect of hyperparameters of IQL on MA2QL, we conduct additional experiments in simple spread with different learning rates and in HalfCheetah with different batch sizes. As shown in Figure 6(b) and Figure 6(c), under these different hyperparameters, the performance of IQL and MA2QL varies, but MA2QL consistently outperforms IQL, which can be evidence of the gain of MA2QL over IQL is insensitive to the hyperparameters of IQL. The default learning rate in MPE is 0.0005 and the default batch size in multi-agent MuJoCo is 100.

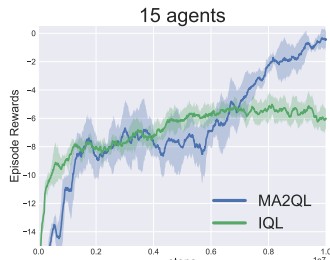

Figure 7: Learning curve of MA2QL compared with IQL in 15-agent simple spread in MPE.

As for scalability, We demonstrate the result of 15-agent simple spread in MPE. As illustrated in Figure 7, MA2QL brings large performance gains over IQL. More agents mean the environments become more complex and unstable. IQL is easy to get stuck by the non-stationarity problem while MA2QL can handle it well. The results show again that alternate learning of MA2QL can sufficiently stabilize the environment during the learning process. It also indicates the good scalability of MA2QL.

## 6   Conclusion and Discussion

In the paper, we propose MA2QL, a simple yet effective value-based fully decentralized cooperative MARL algorithm. MA2QL is theoretically grounded and requires minimal changes to independent Q-learning. Empirically, we verify MA2QL in a variety of cooperative multi-agent tasks, including a cooperative stochastic game, MPE, multi-agent MuJoco, and SMAC. The results show that, in spite of such minimal changes, MA2QL outperforms IQL in both discrete and continuous action spaces, fully and partially observable environments.

MA2QL makes minimal changes to IQL but indeed improves IQL. In practice, MA2QL can be easily realized by letting agents follow a pre-defined schedule for learning. MA2QL also has the convergence guarantee, yet is limited to Nash equilibrium in tabular cases. As a Dec-POMDP usually has many Nash equilibria, the converged performance of MA2QL may not be optimal as shown in the stochastic game. Nevertheless, learning the optimal joint policy in fully decentralized settings is still an open problem. In the stochastic game, we see that IQL also converges, though much slower than MA2QL. This indicates that IQL may also have the convergence guarantee under some conditions, which however is not well understood. We believe fully decentralized learning for cooperative MARL is an important and open research area. However, much less attention has been paid to decentralized learning than centralized training with decentralized execution. This work may provide some insights to further studies of decentralized learning.

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

# A    DETAILED PROOFS FOR THEORETICAL RESULTS

## A.1    PROOF FOR LEMMA 2

*Proof.* From the definition of $Q_i^{t,k}$ (6), we have

$$
\begin{aligned}
\left\|Q_i^{t+1,k} - Q_i^{t,k}\right\|_\infty &= \left\|\gamma\mathbb{E}_{s'\sim P_i^k}[\max_{a_i'} Q_i^{t,k}(s',a_i') - \max_{a_i'} Q_i^{t-1,k}(s',a_i')]\right\|_\infty \\
&\leq \gamma\left\|Q_i^{t,k} - Q_i^{t-1,k}\right\|_\infty \leq \gamma^t\left\|Q_i^{1,k} - Q_i^{0,k}\right\|_\infty.
\end{aligned}
\tag{8}
$$

Then for any integer $m \geq 1$, we have

$$
\begin{aligned}
\left\|Q_i^{t+m,k} - Q_i^{t,k}\right\|_\infty &\leq \left\|Q_i^{t+m,k} - Q_i^{t+m-1,k}\right\|_\infty + \cdots + \left\|Q_i^{t+1,k} - Q_i^{t,k}\right\|_\infty \\
&\leq \gamma^t\frac{1-\gamma^m}{1-\gamma}\left\|Q_i^{1,k} - Q_i^{0,k}\right\|_\infty.
\end{aligned}
\tag{9}
$$

Let $m \to \infty$, and we have

$$
\begin{aligned}
\left\|Q_i^{*,k} - Q_i^{t,k}\right\|_\infty &\leq \frac{\gamma^t}{1-\gamma}\left\|Q_i^{1,k} - Q_i^{0,k}\right\|_\infty \\
&\leq \frac{\gamma^t}{1-\gamma}\max_{s,a_i}\left|r_i^k(s,a_i) + \gamma\mathbb{E}_{s'\sim P_i^k}[\max_{a_i'} Q_i^{0,k}(s',a_i')] - Q_i^{0,k}(s,a_i)\right|.
\end{aligned}
\tag{10}
$$

*As all agents update by turns*, we have $Q_i^{0,k} = Q_i^{t_i^{k-1},k-1}$, where $t_i^{k-1}$ is the number of Q-iteration for agent $i$ in the $k-1$ round. Therefore, we have

$$
\left\|Q_i^{0,k} - Q_i^{*,k-1}\right\|_\infty = \left\|Q_{i-1}^{t_i^{k-1},k-1} - Q_i^{*,k-1}\right\|_\infty \leq \varepsilon.
\tag{11}
$$

With this property, we have

$$
\begin{aligned}
&\left|r_i^k(s,a_i) + \gamma\mathbb{E}_{s'\sim P_i^k}[\max_{a_i'} Q_i^{0,k}(s',a_i')] - Q_i^{0,k}(s,a_i)\right| \\
&= \left|r_i^k(s,a_i) + \gamma\mathbb{E}_{s'\sim P_i^k}[\max_{a_i'} Q_i^{0,k}(s',a_i')] - Q_i^{*,k-1}(s,a_i) + Q_i^{*,k-1}(s,a_i) - Q_i^{0,k}(s,a_i)\right| \\
&\leq \left|r_i^k - r_i^{k-1}\right| + \gamma\left|\mathbb{E}_{s'\sim P_i^k}[\max_{a_i'} Q_i^{0,k}(s',a_i')] - \mathbb{E}_{s'\sim P_i^{k-1}}[\max_{a_i'} Q_i^{*,k-1}(s',a_i')]\right| \\
&\quad + \left|Q_i^{*,k-1}(s,a_i) - Q_i^{0,k}(s,a_i)\right| \leq 2r_{\max} + (\frac{2\gamma r_{\max}}{1-\gamma} + \varepsilon) + \varepsilon = 2R + 2\varepsilon,
\end{aligned}
\tag{12}
$$

where the second term in the last inequality is from $\|Q_i^{*,k-1}\|_\infty \leq \frac{r_{\max}}{1-\gamma}$, $\|Q_i^{0,k}\|_\infty \leq \|Q_i^{*,k-1}\|_\infty + \varepsilon$, and (11). Finally, by combining (10) and (12), we have

$$
\left\|Q_i^{*,k} - Q_i^{t,k}\right\|_\infty \leq \frac{\gamma^t}{1-\gamma}(2R + 2\varepsilon).
\tag{13}
$$

We need $\|Q_i^{*,k} - Q_i^{t,k}\|_\infty \leq \varepsilon$, which can be guaranteed by $t \geq \frac{\log((1-\gamma)\varepsilon) - \log(2R+2\varepsilon)}{\log\gamma}$.    $\square$

## A.2    PROOF FOR THEOREM 1

*Proof.* First, from Lemma 1, we know $Q_i^{*,k}$ also induces a joint policy improvement, thus $Q_i^{*,k}$ converges to $Q_i^*$. Let $\pi_i^*(s) = \arg\max_{a_i} Q_i^*(s,a_i)$, then $\boldsymbol{\pi}^*$ is the joint policy of a Nash equilibrium.

Then, we define $\Delta$ as

$$
\Delta = \min_{s,i}\max_{a_i\neq\pi_i^*(s)}|Q_i^*(s,\pi_i^*(s)) - Q_i^*(s,a_i)|.
\tag{14}
$$

From the assumption we know that $\Delta > 0$. We take $\varepsilon = \frac{\Delta}{6}$, and from Lemma 2, we know there exists $k_0$ such that

$$
\left\|Q_i^* - Q_i^{*,k}\right\|_\infty \leq \varepsilon \quad \forall k \geq k_0.
\tag{15}
$$

For $k \geq k_0$ and any action $a_i \neq \pi_i^*(s)$, we have

$$
\begin{aligned}
Q_i^k(s, & \pi_i^*(s)) - Q_i^k(s, a_i) \\
={} & Q_i^k(s, \pi_i^*(s)) - Q_i^{*,k}(s, \pi_i^*(s)) + Q_i^{*,k}(s, \pi_i^*(s)) - Q_i^*(s, \pi_i^*(s)) \\
& + Q_i^*(s, \pi_i^*(s)) - Q_i^*(s, a_i) + Q_i^*(s, a_i) - Q_i^{*,k}(s, a_i) + Q_i^{*,k}(s, a_i) - Q_i^k(s, a_i) \\
\geq{} & Q_i^*(s, \pi_i^*(s)) - Q_i^*(s, a_i) - |Q_i^k(s, a_i) - Q_i^{*,k}(s, a_i)| - |Q_i^{*,k}(s, a_i) - Q_i^*(s, a_i)| \\
& - |Q_i^*(s, \pi_i^*(s)) - Q_i^{*,k}(s, \pi_i^*(s))| - |Q_i^{*,k}(s, \pi_i^*(s)) - Q_i^k(s, \pi_i^*(s))| \\
={} & \Delta - 4\varepsilon = \Delta/3 > 0,
\end{aligned}
\tag{16}
$$

which means $\pi_i^k(s) = \arg\max_{a_i} Q_i^k(s, a_i) = \arg\max_{a_i} Q_i^*(s, a_i) = \pi_i^*(s)$. Thus, $Q_i^k$ of each agent $i$ induces $\pi_i^*$ and all together induce $\boldsymbol{\pi}^*$, which the joint policy of a Nash equilibrium. $\square$

## B EXPERIMENTAL SETTINGS

### B.1 MPE

The three tasks are built upon the origin MPE (Lowe et al., 2017) (MIT license) and Agarwal et al. (2020) (MIT license) with sparse reward, more agents, and larger space. These tasks are more difficult than the original ones. In the original tasks, we found IQL and MA2QL perform equally well, which corroborates the good performance of IQL on simple MPE tasks (Papoudakis et al., 2021). Specifically, we change the dense reward setting to the sparse reward setting that will be described below. We increase the number of agents from 3 to 5 in the task of simple spread and enlarge the range of initial positions from [-1, 1] to [-2, 2]. A larger space would require the agents to explore more and the agents would be easier to get stuck in sub-optimum. As there is no distance signal in the sparse reward setting, we add a boundary penalty into the environment. If the agents move outside the boundaries, the environment would return the negative reward of -1. The boundaries in our experiment are [-3, 3].

**Simple spread.** In this task, the agents need to find some landmarks in a 2D continuous space. In the sparse reward setting, only if the agents cover the landmarks, the environment would return the positive reward. There is no reward after the landmarks are covered. Whether the landmark is covered by the agent is based on the distance between the landmark and the agent. We set a threshold of 0.3 in our experiment. If the distance is smaller than the threshold, we consider the landmark is covered by the agent. We set 5 agents and 4 landmarks in this task. Fewer landmarks require the agents to do more exploration in the space to find the landmarks.

**Line control.** In this task, there are 5 agents in the space. The goal of the task is that the agents need to arrange themselves in a straight line. The reward depends on the number of agents in the straight line. If the straight line is formed by the agents of 5, the environment would return the reward of 5. Since the action space is discrete, we set a threshold of 15° to judge whether it is a line.

**Circle control.** There are 7 agents and 1 landmark in the space. This task is similar to the task of line control, while the agents are asked to form a circle where the landmark is the center. We also set a threshold of 0.15.

### B.2 MULTI-AGENT MUJOCO

Multi-agent MuJoCo (Peng et al., 2021) (Apache-2.0 license) is built upon the single-agent MuJoCo (Todorov et al., 2012). A robot can be considered as the combination of joints and body segments. These joints and body segments in a robot can be regarded as the vertices and connected edges in a graph. We can divide a graph into several disjoint sub-graphs and a sub-graph can be considered as an agent. We can design new observation and action spaces for each agent based on the division. For example, the robot of Walker2d has 6 joints in total. We can divide joints and body segments into left parts and right parts as two agents.

The details about our experiment settings in multi-agent Mujoco are listed in Table 1. The configuration defines the number of agents and the joints of each agent. The "agent obsk" defines the number of nearest agents an agent can observe. Thus, HalfCheetah and Hopper are fully observable, while Walker2D is partially observable.

Table 1: The environment settings of multi-agent MuJoCo

| task | configuration | agent obsk |
|------|--------------|-----------|
| HalfCheetah | $2 \times 3$ | 1 |
| Hopper | $3 \times 1$ | 2 |
| Walker2d | $3 \times 2$ | 1 |

## C   TRAINING DETAILS

### C.1   MPE

We use DRQN with an RNN layer and MLP as the architecture of Q-network. The RNN layer is composed of a GRU with the 64-dimension hidden state. We utilize ReLU non-linearities. All networks are trained with a batch size of 128 and Adam optimizer with the learning rate of 0.0005. There are 16 parallel environments that collect transitions and the replay buffer with the size of 5000 contains trajectories. For exploration, we use $\epsilon$-greedy policy and $\epsilon$ starts from 1.0 and decreases linearly to 0.1 in the first 50000 timesteps.

For MA2QL, we define $K$ is the training steps for updating an agent at each turn. For instance, it is the turn to update the agent $i$. After training $K$ steps, it would change the turn to update the agent $i + 1$. We set $K$ as 30 in this experiment.

### C.2   MULTI-AGENT MUJOCO

We use DDPG for continuous action space and the setting is similar to the one used in OpenAI Spinning Up (Achiam, 2018) (MIT license). The architecture of both actor network and critic network is MLP with two hidden layers with 256-dimension hidden state. We utilize ReLU non-linearities except the final output layer of the actor network. The activation function of the final output layer of the actor network is tanh function that can bound the actions. All networks are trained with a batch size of 100 and Adam optimizer with the learning rate of 0.001. There is only one environment that collects transitions and the replay buffer contains the most recent $10^6$ transitions. For exploration, the uncorrelated, mean-zero Gaussian noise with $\sigma = 0.1$ is used. For MA2QL, we set $K$ as 8000. Unlike other two environments where agents learn on trajectories, in multi-agent MuJoCo, agents learns on experiences. Thus, in each turn, the agent can be updated every environment step and hence $K$ is set to be large.

### C.3   SMAC

Our code is based on the Hu et al. (2021) (Apache-2.0 license) and the version of SMAC is SC2.4.10. The architecture of Q-network is DRQN with an RNN layer and MLP. The RNN layer is composed of a GRU with the 64-dimension hidden state. All networks are trained with a batch size of 32 and Adam optimizer with the learning rate of 0.0005. All target networks are updated every 200 training steps. There are 2 parallel environments that collect transitions and the replay buffer with the size of 5000 contains trajectories. For exploration, we use $\epsilon$-greedy policy and $\epsilon$ starts from 1.0 and decreases linearly to 0.05 in the first 100000 timesteps. We add the action of last timestep into the observation as the input to the network. For MA2QL, we set $K$ as 50.

### C.4   RESOURCES

We perform the whole experiment with a total of ten Tesla V100 GPUs.

## D   ADDITIONAL RESULTS

### D.1   ADDITIONAL SMAC RESULTS

We also demonstrate the results on more maps in SMAC. As shown in Figure 8, IQL shows strong performance in these maps, which corroborates the good performance of IQL in SMAC (Papoudakis

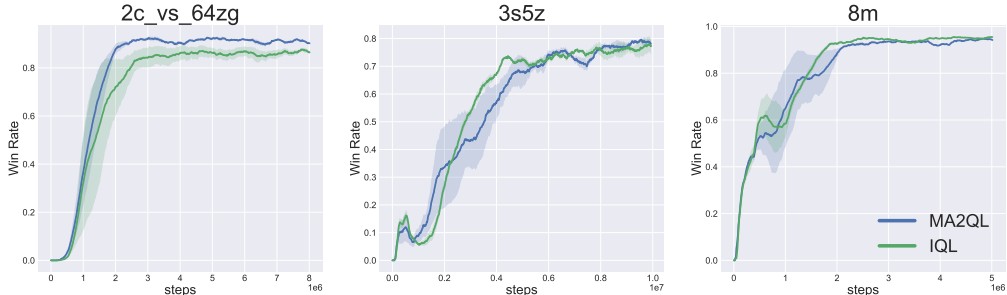

Figure 8: Learning curve of MA2QL compared with IQL on `2c_vs_64zg` (hard), `3s5v` (hard) and `8m` (easy) in SMAC, where x-axis is environment steps.

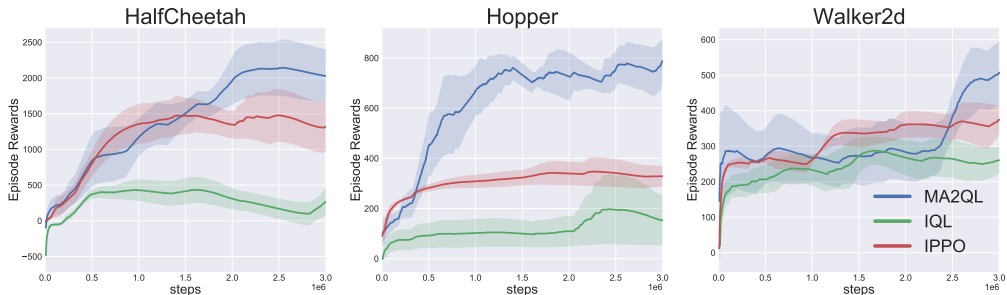

Figure 9: Learning curve of MA2QL compared with IQL and IPPO in 2-agent HalfCheetah, 3-agent Hopper and 3-agent Walker2d in multi-agent MuJoCo, where x-axis is environment steps.

et al., 2021). MA2QL performs similarly to IQL on the maps of `3s5v` and `8m`, but better on the map `2c_vs_64zg`. Together with the results in Section 5.4, we can see that although IQL performs well in SMAC, MA2QL can still bring performance gain in more difficult maps, which again verifies the superiority of MA2QL over IQL.

## D.2 ADDITIONAL COMPARISON WITH IPPO IN MULTI-AGENT MUJOCO

As MA2QL is derived from Q-learning and thus a value-based method, we focus on the comparison with IQL. Here we provide the results additionally compared with an actor-critic method, *i.e.*, independent PPO (IPPO) (de Witt et al., 2020), in the experiment of multi-agent MuJoCo. The setting and the architecture of IPPO are also similar to the one used in OpenAI Spinning Up (Achiam, 2018) (MIT license). As illustrated in Figure 9, MA2QL also consistently outperforms IPPO.

## D.3 ALTERNATE UPDATE OF IPPO

The principle of alternate update can also be applied to independent actor-critic methods. Here we additionally provide the empirical results of alternate update of IPPO (termed as MA2PPO) in the cooperative stochastic game and multi-agent MujoCo. For the cooperative stochastic game, as illustrated in Figure 10, MA2PPO substantially outperforms IPPO. For multi-agent MuJoCo, as shown in Figure 12, MA2PPO brings performance gain in HalfCheetah and performs comparably with IPPO in Hopper and Walker2D. These results may suggest that such a simple principle can also be applied to independent actor-critic methods for better performance. A more thorough investigation on independent actor-critic methods is beyond the scope of this paper and left as future work.

## D.4 ALTERNATE UPDATE IN CTDE

Though the principle of alternate update can still be applied to CTDE methods, there may not be many benefits. In the CTDE setting, there are actually a number of algorithms that have convergence guarantee. Without the advantage of convergence, alternate update will actually make the learning

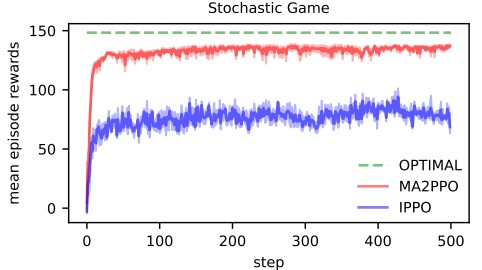 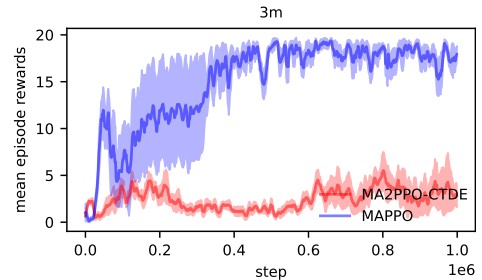

Figure 10: Learning curves of MA2PPO compared with IPPO in the cooperative stochastic game.

Figure 11: Learning curves of MA2PPO-CTDE compared with MAPPO on the map 3m in SMAC.

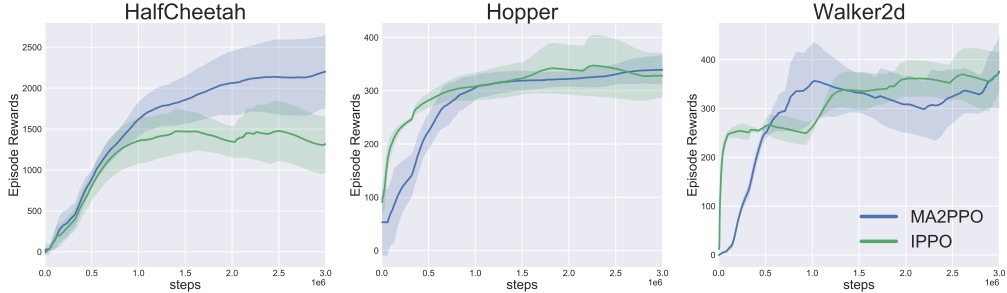

Figure 12: Learning curves of MA2PPO compared with IPPO in 2-agent HalfCheetah, 3-agent Hopper and 3-agent Walker2d in multi-agent MuJoCo, where x-axis is environment steps.

procedure slower. We modified the CTDE algorithm MAPPO (Yu et al., 2021) with alternate update (termed as MA2PPO-CTDE) and compare it with MAPPO in the simple map 3m in SMAC. The empirical result is illustrated in Figure 11. We can find that MA2PPO-CTDE obtains low performance while MAPPO has the performance which is close to the optimum. Though alternate update may become better after a longer training, alternate update is not quite meaningful in CTDE.

## E    ADDITIONAL RELATED WORK

Our method shares the same principle of alternate update with existing studies for optimization, *e.g.*, alternating gradient descent (Lu et al., 2019) and coordinate gradient descent (Diakonikolas & Orecchia, 2018). There is also existing work (Fang et al., 2019; Bertsekas, 2020) investigating alternate update for multi-agent reinforcement learning, which however is only limited to the theoretical result of Lemma 1 in Section 2.

## F    ADDITIONAL EXPERIMENTS FOR REBUTTAL

We carried out additional experiments in the didactic game, including different stochasticity levels, exploration schemes, and update orders.

First, we control the stochasticity level of the environment by introducing $p\_trans$. For any transition, a state has the probability of $p\_trans$ to transition to next state uniformly at random, otherwise follows the original transition probability. Thus, larger $p\_trans$ means higher level of stochasticity. As illustrated in Figure 13, MA2QL outperforms IQL in all stochasticity levels.

We also investigate the performance of MA2QL under different exploration schemes, including different decaying rates and constant $\epsilon$. As illustrated in Figure 14, MA2QL again outperforms IQL in all these exploration schemes. Note that $decay\_100K$ means $\epsilon$ decays from 1 to 0.02 over 100K environment steps, similarly for others. These results shows the robustness of MA2QL under various exploration schemes, even when $\epsilon = 1$.

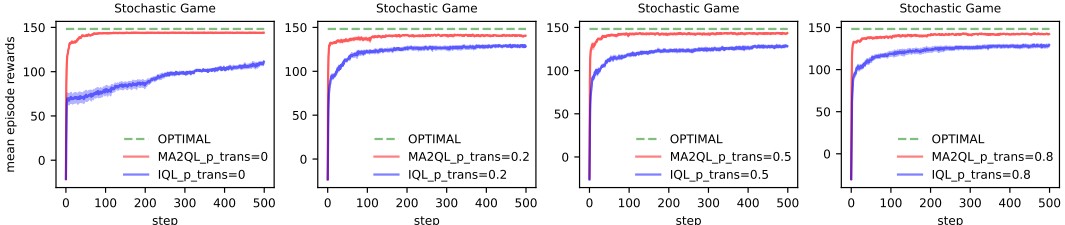

Figure 13: Learning curves of MA2QL compared with IQL in the didactic game under different stochasticity levels, where $p\_trans$ is the probability of a state transitions to next state uniformly at random and hence larger $p\_trans$ means higher level of stochasticity.

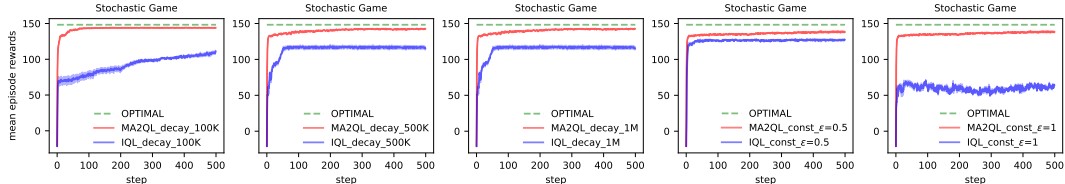

Figure 14: Learning curves of MA2QL compared with IQL in the didactic game under different exploration schemes, where $decay\_100K$ means $\epsilon$ decays from 1 to 0.02 over 100K environment steps, similarly for others.

We further investigate the performance of MA2QL under different pre-defined update orders and random order at each round. As there are three agents in the environment, there are essentially two different pre-defined orders. Suppose that three agents are indexed from 0 to 2. The alternating update orders [0,1,2], [1,2,0], and [2,0,1] are the same, while [0,2,1], [1,0,2], and [2,1,0] are the same. As illustrated in Figure 15, the performance of MA2QL is almost the same under the two orders (MA2QL is the order of [0,1,2]), which shows the robustness of MA2QL under different pre-defined orders. As for random order at each round, the performance drops but is still better than IQL. One possible reason is that agents are not evenly updated due to the random order at each round, which may consequently induce a worse Nash equilibrium. However, a pre-defined order is better and more practical than random order that requires all agents reach an agreement of who should update at each round.

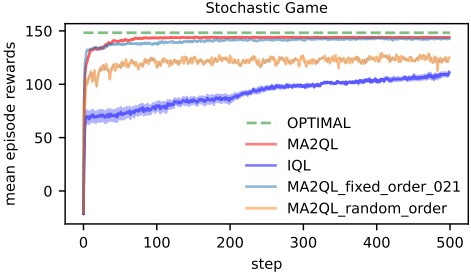

Figure 15: Learning curves of MA2QL of different update orders. There are three agents in the didactic game, so there are essentially two different pre-defined update orders: [0,1,2] (MA2QL) and [0,2,1].

We also carried out the experiments in a matrix game, of which the payoff matrix is illustrated in Figure 16(a). The learning curves of MA2QL and IQL are shown in Figure 16(b), where left is linearly decayed $\epsilon$ and right is $\epsilon = 1$. MA2QL and MA2QL with $\epsilon = 1$ both learn the optimal joint action $AA$ while IQL learn the suboptimal joint action $CC$. For both MA2QL and MA2QL with $\epsilon = 1$, the learned Q-table of two agents are the same, which is $[11, -30, 0]$ for action $[A, B, C]$ respectively. For IQL, the Q-table of $a_1$ is $[-2.2, 1.9, 6.9]$, and the Q-table of $a_2$ is $[-2.1, -2.7, 6.9]$. For IQL with $\epsilon = 1$, the Q-table of $a_1$ is $[-6.3, -5.6, 2.3]$, the Q-table of $a_2$ is $[-6.3, -7.7, 4.3]$. Although the learned joint action in both cases is $CC$, the value estimate of $CC$ is less accurate for IQL with $\epsilon = 1$. Meanwhile, MA2QL has the same accurate value estimate regardless of exploration,

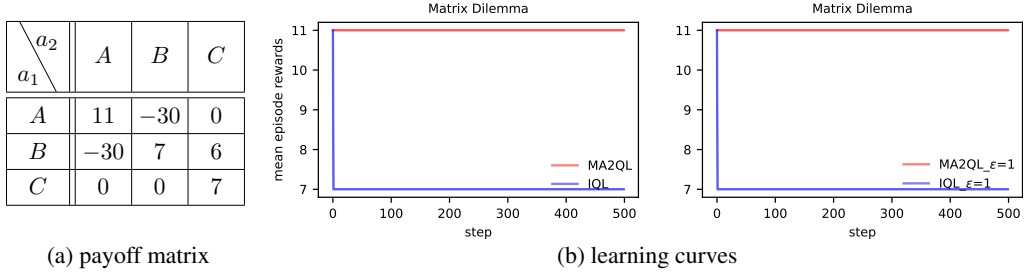

(a) payoff matrix           (b) learning curves

Figure 16: A matrix game: (a) payoff matrix; (b) learning curves of MA2QL and IQL with linearly decayed $\epsilon$ (left) and $\epsilon = 1$ (right).

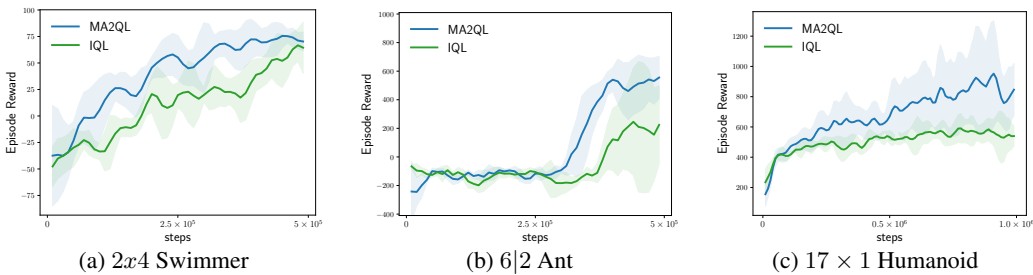

(a) $2x4$ Swimmer      (b) $6|2$ Ant      (c) $17 \times 1$ Humanoid

Figure 17: Learning curve of MA2QL compared with IQL in 2-agent Swimmer, 6-agent Ant, and 17-agent Humanoid in multi-agent MuJoCo, where x-axis is environment steps.

which corroborates the good performance of MA2QL under different exploration schemes in the didactic game.

Moreover, we carried out additional experiments in multi-agent MuJoCo, including $2 \times 4$ Swimmer (agent_obsk=0), $6|2$ Ant (agent_obsk=0), and $17 \times 1$ Humanoid (full observation). The results are shown in Figure 17, and MA2QL also outperforms IQL in these tasks.

