# OpenReview forum: "MA2QL: A Minimalist Approach to Fully Decentralized Multi-Agent Reinforcement Learning"
_ICLR.cc/2023/Conference — Submitted to ICLR 2023_

### Official Review · Reviewer_xmYF · 2022-10-22

**Confidence:** 4
**Correctness:** 3
**Technical Novelty And Significance:** 2
**Empirical Novelty And Significance:** 2
**Recommendation:** 3

**Clarity, Quality, Novelty And Reproducibility:**

The clarity of this paper is good, both theoretical analysis and experimental analysis are clear.

As for the originality, although this idea should be independently raised from the authors, the similar ones have appeared so many times in the past. To me, the theoretical result shown in this paper is unsurprising and old.

The reproducibility is good, since the authors provide the experimental settings.

Overall, the quality of this paper is good from the presentation, but not good enough from the contents.

**Strength And Weaknesses:**

## Strength
1. The motivation of this paper is clear and the discussion of the related works are sufficient.
2. The theoretical analysis of this paper is easy to follow and correct.
3. The experiment design is comparatively good which covers variants of scenarios.
4. The proposed algorithm is neat.

## Weaknesses
1. Simultaneous move (update) and iterative move (update) are two typical paradigms existing in both game theory and multi-agent learning. Iterative move (update), e.g. the original version of Brown's fictitious play, has been proved to converge to Nash equilirbium in potential game. The team reward game considered in this paper is a special case of the potential game. Even equipped with Markov dynamics, there is also some existing work to show the same idea as the paper proposes [1]. In addition, in the recent paper [2] the alternate update (or rollout in sequence) has also been discussed. Therefore, the theoretical contribution of this paper is minor in my view. From the multi-agent learning side, there is a past paper [3] that also shares the similar idea.
2. As for the practical aspect, the proposed MA2QL will increase the times of the interaction with the environment to n when there are n agents. It will lead to extra cost as the number of agents increases. It is meaningless to practice in my view, unless it can exceed the state-of-the-art methods largely.


## Reference
[1] Hu, Ruimeng. "Deep fictitious play for stochastic differential games." arXiv preprint arXiv:1903.09376 (2019).

[2] Bertsekas, Dimitri. "Multiagent reinforcement learning: Rollout and policy iteration." IEEE/CAA Journal of Automatica Sinica 8.2 (2021): 249-272.

[3] Lauer, Martin, and Martin Riedmiller. "An algorithm for distributed reinforcement learning in cooperative multi-agent systems." In Proceedings of the Seventeenth International Conference on Machine Learning. 2000.


**Summary Of The Paper:**

This paper proposes a modification to IQL called MA2QL in which each agent alternately updates the parameters rather than the simultaneous update. The main contribution of this paper is the theoretical analysis of this modification. In experiments, MA2QL is demonstrated to outperform IQL in several environments.

**Summary Of The Review:**

This paper proposes a simple modification of IQL and give a complete theoretical analysis to show the convergence property. However, the similar theoretical results have appeared in the past several times, so the novelty of this paper faces a challenge. From the practical side, the proposed alternative udpate will lead to n times interactions with the environment which is a huge cost as the number of agents increases.

---

> ### Author Response · Authors · 2022-11-15
> **Response**
>
> > related work and novelty
>
> Thanks for bringing up these works. As mentioned in Appendix E,  we indeed noticed some works, e.g., [4][5], sharing the same idea. However, for [1][4][5], **their theoretical results are limited to Lemma 1 in Section 2**. That is the reason why we treat Lemma 1 as background. The proposed method in [2], from our understanding, is similar to applying alternate update to PPO, termed MA2PPO in Appendix D. However, this is not the focus of our paper.
>
> The main theoretical result of this paper is Theorem 1, which lays the theoretical ground that the convergence of Q-function at each round is not necessary. Based on this insight, we designed MA2QL, which outperforms IQL in a variety of tasks empirically. This is the main contribution of our paper.
>
> Our work is different from [3]. In [3], all agents learn simultaneously, **but limited to deterministic environments**. MA2QL does not have this limitation.
>
> [1] Hu, Ruimeng. Deep fictitious play for stochastic differential games. arXiv:1903.09376.
>
> [2] Bertsekas, Dimitri. Multiagent reinforcement learning: Rollout and policy iteration. IEEE/CAA Journal of Automatica Sinica 8.2 (2021): 249-272.
>
> [3] Lauer, Martin, and Martin Riedmiller. An algorithm for distributed reinforcement learning in cooperative multi-agent systems. ICML 2000.
>
> [4] Dimitri Bertsekas. Multiagent value iteration algorithms in dynamic programming and reinforcement learning.  Results in Control and Optimization, 1:100003, 2020.
>
> [5] Fang et al., Multi-agent cooperative alternating q-learning caching in d2d-enabled cellular networks. In IEEE Global Communications Conference 2019.
>
> > increased number of interactions
>
> Please do note that for all the experiments, we compared MA2QL with IQL using the same number of interactions (i.e., environment steps). MA2QL outperforms IQL, which verifies the benefit of our method.  That said, MA2QL does not lead to an increased number of interactions.

---

> ### Author Response · Authors · 2022-11-15
> **Change of Score**
>
> **The score was 5, but now it is 3. The reviewer changed the score before we posted the response. Is it professional?**

---

### Official Review · Reviewer_p9BU · 2022-10-23

**Confidence:** 4
**Correctness:** 3
**Technical Novelty And Significance:** 2
**Empirical Novelty And Significance:** 2
**Recommendation:** 3

**Clarity, Quality, Novelty And Reproducibility:**

The paper has good clarity overall, although the definition of the specific problem setting considered is not clear to me (and this is one major problem). Regarding novelty, I think the main idea of "fixing the policies of other agents while one agent is learning" in MA2QL is not new. Very similar idea has been used in previous works on competitive MARL [1].

**Strength And Weaknesses:**

- Strengths:
	- The main approach proposed is theoretically grounded, with some good experimental results.
	- The paper is well-written and easy to follow overall.
- Weaknesses:
	- The specific "fully decentralized" multi-agent setting considered (where parameter sharing is not allowed) in this work is neither well motivated nor clearly explained, which is critical for justifying the significance of this work.
	- The main idea of  the proposed algorithm (i.e., "fixing the policies of other agents while one agent is learning") is not new. For instance, it has been used in existing works in competitive MARL [1].
	- The experiments are not sufficiently comprehensive to me (e.g., MA2QL is only compared against IQL).

**Summary Of The Paper:**

This paper proposes multi-agent alternate Q-learning (MA2QL) to tackle the non-stationarity problem in decentralized MARL. The main difference between MA2QL and existing method IQL is that MA2QL agents take turns to update their Q-functions while IQL agents simultaneously update their Q-functions. The authors prove that, in MA2QL, when each agent guarantees $\epsilon$-convergence at each turn, their joint policy converges to a Nash equilibrium. They empirically show that MA2QL outperforms IQL in a variety of cooperative multi-agent tasks with discrete and continuous action spaces.

**Summary Of The Review:**

My biggest concern of this paper is the specific "fully decentralized" multi-agent setting considered (where parameter sharing is not allowed) is not well motivated and clearly explained. It is not clear to me what the practical use cases for this setting are. I can think of some real-world scenarios where there are separate machines/robots who are not able to/not allowed to share parameters with each other. But in MA2QL, you seem to be implicitly assuming some kind of synchronization mechanism that enables the agents to reach an agreement on who should update their policy/Q-function at each learning stage. In that sense, is it really "fully decentralized" based on your definition?

My another main concern is the idea introduced in MA2QL is not really new as I've mentioned above. In addition, for the experiments, while I appreciate the evaluation of MA2QL on a variety of cooperative multi-agent tasks with discrete and continuous action spaces (and full/partial observability), I think comparing only against IQL is not enough. What about other independent learning baselines such as independent actor-critic, independent DDPG, and independent PPO? For instance, does MA2QL outperform independent PPO in SMAC?

Here are some other comments/questions:
- "we do not use parameter-sharing, which should not be allowed in decentralized settings (Terry et al., 2020)." I do not understand why parameter sharing "should not be allowed in decentralized settings."
- In Multi-Agent MuJoCo, using names MA2QL and IQL for the algorithms while DDPG is actually used are misleading.
- In MA2QL, how do you decide the order in which each agent updates their Q-functions? Is it a random order? Do agents need to fully agree on who should update their Q-functions next?
- For the hyperparameter K used in MA2QL, it is set to be 30 in MPE but 8000 in Multi-Agent MuJoCo. The range of values looks quite large. How do you efficiently tune this hyperparameter?

[1] Lanctot, Marc, Vinicius Zambaldi, Audrunas Gruslys, Angeliki Lazaridou, Karl Tuyls, Julien Pérolat, David Silver, and Thore Graepel. A unified game-theoretic approach to multiagent reinforcement learning. In Advances in Neural Information Processing Systems, 2017.

---

> ### Author Response · Authors · 2022-11-15
> **Response**
>
> > about fully decentralized
>
> As commented by Reviewer SmxM in "strengths",  fully decentralized learning is useful in real-world learning scenarios where no good simulator is available or communication during training is strictly limited, This should address your concern. Please be open-minded, not all problems can be solved by CTDE.
>
> Besides, MA2QL needs a pre-defined random order, which however is not difficult in practice.
>
>
> > about the related work [1]
>
> Lanctot et al. [1] aim to learn a population of policies for better generalization in multi-agent settings, where each oracle is learned by fixing other players. So, the problem considered in [1] is quite different from ours.
>
> Moreover, as we mentioned in Appendix E, several previous works share the same principle, however, they are all limited to Lemma 1 in Section 2. The main theoretical result of this paper is Theorem 1, which lays the theoretical ground that the convergence of Q-function at each round is not necessary.
>
> [1] Lanctot et al., A unified game-theoretic approach to multiagent reinforcement learning. NeurIPS 2017.
>
> > about experiments and baselines
>
> In multi-agent MuJoCo, IQL is indeed independent DDPG. We also compared MA2QL with IPPO in MuJoCo in Appendix. We also applied the principle of MA2QL, i.e., alternate learning, to IPPO in the didactic game, and it is shown that it also outperforms IPPO. As stated in the second paragraph in Section 4, as actor-critic methods (e.g., PPO) follow the principle different from Q-learning, we do not focus on independent actor-critic for comparison in the experiment.
>
> > parameter sharing
>
> To share parameters, we have to have a centralized entity to collect all parameters (at least full communication), which contradicts the setting of decentralized learning.
>
> > update order
>
> Theorem 1 holds in any update order.  So, in the experiments, we use a pre-defined order. We additionally perform experiments using different pre-defined orders in the didactic game.  The results are included in Appendix F. For the pre-defined orders, their performance is almost the same. For random order at each round, MA2QL still outperforms IQL.
>
> > about hyperparameter K
>
> As mentioned in Appendix C, in MPE and SMAC, the network is based RNN. So, the update can only be performed on episode, while in MuJoCo, the update is performed on environment step. This is the reason why they are quite different. The effect of hyperparameter K is investigated in Figure 6 (a).

---

> ### Author Response · Authors · 2022-11-18
> **Follow-up**
>
> Please let us know whether our responses have addressed your concerns. Also, we are willing to address any additional comments.
>
> Thanks.

---

### Official Review · Reviewer_SmxM · 2022-10-24

**Confidence:** 4
**Correctness:** 4
**Technical Novelty And Significance:** 3
**Empirical Novelty And Significance:** 3
**Recommendation:** 6

**Clarity, Quality, Novelty And Reproducibility:**

**Questions:**

- Regarding this statement: "*One may notice that the performance of MA2QL is better than MA2QL-DP. This may be attributed to sampling and exploration of MA2QL, which induces a better Nash equilibrium.*"

   1. How did you perform DP updates? Full sweeps over the entire set of states in every update? Or were the backup states sampled according to, e.g., the buffered transitions?

   2. I cannot fully understand what you exactly mean in the statement above. Could you elaborate?

- "*To investigate continuous action space in both partially and fully observable environments, we configure 2-agent HalfCheetah and 3-agent Hopper as fully observable, and 3-agent Walker2d as partially observable.*"

   - Could you please add results for the fully-observable case with a higher number of agents in this benchmark?

- "*Here, we show that MA2QL can still outperform IQL in three maps with various difficulties, which indicates that MA2QL can
also tackle the non-stationarity problem and bring performance gain in more complex tasks.*"

   - Is there additional non-stationarity in SMAC tasks beyond the non-stationarity that exists in general for IQL?

- Do any of these benchmarks have stochastic transition dynamics and stochastic reward functions? A big problem with non-stationarity in MARL is when there is stochasticity in the environment, whereby the challenge is partly that the agents cannot distinguish between the stochasticity in the environment from that induced due to changing policies of other cooperative teammates. I believe some experiments with the stochastic version of these environments, or at least a subset of the tasks are required. Also, it would even be better if one experiment compares IQL vs MA2QL with varying stochasticity levels in the environment.

- What is the exploration strategy schedule? Are there any experiments with different exploration schemes? For e.g. what if we use IQL against MA2QL on a task with linearly-decayed epsilon of varying decay rates and with varying fixed-epsilon/final-epsilon levels? If MA2QL shows more robustness across varying exploration schemes, then it would highly support the main arguments of the paper.

- Could you also test IQL vs. MA2QL on a tabular dilemma problem concerning "relative overgeneralization (or shadowed equilibria)"? E.g., consider the following 2-agent tabular dilemma task with the reward function:
```
 11  |  -30   |  0
-30  |   7    |  6
 0   |   0    |  7
```
I'm curious what would be the learned Q-tables for the agents and the implied policy over the joint action space for MA2QL using $\epsilon$-greedy exploration with epsilon=1.0.

- Is there any intuition/results that could give an idea of whether a similar scheme could also help improve VDN (or Value Decomposition Networks)? In the appendix, there is some argument that MAPPO (CTDE) does not benefit from Alternate Updates (as in MA2PPO). I'm not quite sure why this is, and I also would've really preferred seeing this in the context of VDN.

- What is "the CTDE algorithm MAPPO"? I cannot easily find any summary of this agent (I don't know what MAPPO does to be CTDE beyond IPPO).

- "*The reason is that their experiences are generated in different manners.*"

This is confusing to me: are the experiences generated with only one agent potentially exploring and others exploiting, or do you mean something else here?

- "*MA2QL obtains experiences with only one agent learning, so the variation for the joint policy is smaller than that of IQL. Thus, in
general, the divergence between $π$ and $π_D$ is smaller for MA2QL, which is beneficial to the learning.*"

Doesn't the same argument benefit training of VDN? Why?

- Regarding Appendix E (Additional Related Works): Could you add more info about what these papers are about, and what do they explore, especially those in the context of MARL?

**Strength And Weaknesses:**

**Strengths:**

- The idea is simple and elegant.

- The idea is more realistic than standard IQL, in which updates have to be synchronized across agents. Their approach MA2QL, on the other hand, does not require synchrony and in fact is closer to what would happen in a practical physical multiagent system, where agents will receive updates at semi-fixed intervals of time without a synchronized starting time.

- Improving Independent Learning is much harder than CTDE, and advances in this direction could prove more useful in real-world learning scenarios where no good simulator is available or communication during training is strictly limited.


**Weaknesses:**

- While the paper is generally well-written, I struggled to obtain answers to many questions: partly due to the way things are explained/described, and partly due to missing analysis/experiments. Please refer to my questions in the next section to help clarify some of these concerns.

- Novelty might be limited with some related work in the context of MARL focusing on the same concept (see Appendix E for some such connections discussed by the authors, but not in great detail). I hope the authors can discuss this more.

**Summary Of The Paper:**

The paper proposes a simple modification to Independent Q-Learning to make it better address the non-stationarity issue in credit assignment. The modification involves allowing agents to take turns in updating their individual Q-tables or Q-functions. This is as opposed to synchronous updates in IQL implementations. This paper motivates this approach through theory (in the tabular value iteration and Q-learning) and experiments (across several standard benchmarks).

**Summary Of The Review:**

I generally like the paper, but I have several concerns that I hope the authors can address in their rebuttal.

---

> ### Author Response · Authors · 2022-11-15
> **Response**
>
> >  about DP update
>
> First, DP is full sweeps over the entire set of states. So, there is no sampling of states. However, MA2QL outperforms MA2QL-DP. Why is that? One possible reason is that the randomness/bias induced by sampling and exploration made MA2QL converge to a better Nash equilibrium than  MA2QL-DP.
>
> > fully-observable case with a larger number of agents in MuJoCo
>
> We additionally perform 17-agent Humanoid (fully observable). MA2QL still outperforms IQL. The result is included in Appendix F.
>
>
> > SMAC and stochastic environment
>
> There is no additional non-stationarity in SMAC tasks beyond the non-stationarity that exists in general for IQL.
>
> SMAC is the benchmark of stochastic transition dynamics (as indicated in the GitHub of SMAC), but the stochasticity level cannot be controlled.
>
> To investigate the performance of MA2QL in varying stochasticity levels, we control the stochasticity in the didactic game. The results are included in Appendix F. Under different stochasticity levels, MA2QL consistently outperforms IQL, which shows the robustness of MA2QL.
>
> > exploration
>
> For MPE and SMAC, the exploration is linearly decayed $\epsilon$. For MuJoCo, the exploration is mean-zero Gaussian noise with $\sigma=0.1$ as indicated in Appendix C.
>
> As suggested, we test more exploration schemes, including different fixed $\epsilon$ and decay rates. The results show that the performance of MA2QL varies very slightly under these schemes and it also consistently outperforms IQL. The results are included in Appendix F.
>
> > 2-agent matrix dilemma with $\epsilon=1$
>
> We compare MA2QL and IQL in this suggested matrix game with $\epsilon=1$. The results show that MA2QL converges to the optimum (11) while IQL converges to 7. The results are included in Appendix F.
>
>
> > about VDN and MAPPO
>
> Unlike IPPO, MAPPO learns a centralized V-function to compute the gradient for each agent.
>
> Applying alternate learning to CTDE methods including VDN and MAPPO is not meaningful.  Alternate learning is used to simply fix the non-stationary problem. As CTDE methods have already addressed non-stationarity and all agents can learn simultaneously, applying alternate learning to CTDE has no benefit and can also impair its learning efficiency.
>
> The argument "MA2QL obtains experiences with only one agent learning....., which is beneficial to the learning" does not benefit the training of VDN. As VDN is an off-policy CTDE method, the smaller divergence between $\pi$ and $\pi_D$ does not necessarily benefit the learning.
>
>  > "The reason is that their experiences are generated in different manners."
>
> Yes, we mean that in MA2QL the experiences are generated with only one agent potentially exploring and others exploiting.
>
> > Related work in Appendix E
>
> The related work including [1][2] in Appendix E and [3] brought by Reviewer xmYF also studies alternate learning in MARL in different contexts. However, their core ideas are the same, i.e., one agent learns using Q-learning till convergence while fixing other agents. However, this is simple, and their theoretical results are summarized by Lemma 1 in Section 2. This is the reason we treat Lemma 1 as **background**.  For [4], the proposed method is similar to MA2PPO in Appendix. However, this is not the focus of our paper.
>
> The main theoretical result of this paper is Theorem 1, which lays the theoretical ground that the convergence of Q-function at each round is **unnecessary**. Based on this insight, we designed MA2QL, which outperforms IQL in a variety of tasks empirically.
>
>
> [1] Fang et al., Multi-agent cooperative alternating q-learning caching in d2d-enabled cellular networks. In IEEE Global Communications Conference 2019.
>
> [2] Dimitri Bertsekas. Multiagent value iteration algorithms in dynamic programming and reinforcement learning.  Results in Control and Optimization, 1:100003, 2020.
>
> [3] Hu, Ruimeng. Deep fictitious play for stochastic differential games. arXiv:1903.09376.
>
> [4] Bertsekas, Dimitri. Multiagent reinforcement learning: Rollout and policy iteration. IEEE/CAA Journal of Automatica Sinica 8.2 (2021): 249-272.

---

> > ### Comment · Reviewer_SmxM · 2022-11-17
> > **Thanks for the additional experiments and your response**
> >
> > Thank you for your responses, which clarified my questions/concerns. I quite enjoyed the new results and analyses that you added after the rebuttal. But please for the final version, try to incorporate them more seamlessly into meaningful sections, rather than putting all results in one section for results after the rebuttal.
> >
> > Regarding novelty, I will have to look closer at the related works and discuss them with other reviewers. However, as per my personal concerns regarding the paper, I believe they have been sufficiently addressed. As such, I have increased my score to reflect that.

---

> > > ### Author Response · Authors · 2022-11-18
> > > **Response**
> > >
> > > It is great that our responses have addressed your questions/concerns.
> > >
> > > We will re-organize the paper to incorporate these new results seamlessly in the next version.
> > >
> > > Thanks for the acknowledgment of our contribution.

---

### Official Review · Reviewer_J8oo · 2022-10-25

**Confidence:** 4
**Correctness:** 3
**Technical Novelty And Significance:** 2
**Empirical Novelty And Significance:** 2
**Recommendation:** 3

**Clarity, Quality, Novelty And Reproducibility:**

The paper is generally easy to read and well-structured. However, more discussions are required to support the effectiveness of alternate learning. Other than that, naively using alternate learning may suppress the novelty. In addition, several experiments should be performed for more clarification. Finally, reproducibility should be improved.

**Strength And Weaknesses:**

While this paper is well-organized, easy to follow, and theoretically supported, there is a fundamental debate on alternate learning concepts which should be clearly explained. In addition, there is a gap between theory and practice which is not properly dealt with. Experiments should be expanded for further analysis, and finally, reproducibility cannot be verified.

Q1. Fundamental discussions on alternate learning

The core concept of this paper is alternate learning. There are substantial issues regarding the alternate learning concept.
- Imagine we consider an extremely large distributed system such as decentralized traffic control. Training each agent alternatingly requires an extreme amount of time and each agent may not adequately take action at each moment.
- If $n$ is extremely big, each agent is not fully trained because the period is long. In fact, in the wall-clock time sense, IQL can conduct more gradient steps within the same wall-clock time to improve performance. In this sense, “a fair comparison” in this paper may be questionable.
- This paper does not explicitly describe the effect of agent order. Can the order be random at each round, or the order should be predefined? Can the authors show any numerical study showing whether order matters or not? If $n$ is extremely large, there can be one good agent and the other bad agents, which can cause asymmetric bias. Then it can be sensitive to the order.
- Though controversial, defining agent order may not correspond to fully decentralized learning (unless every agent agrees to predefined order “protocol” before starting training) since each agent should know whether a certain turn is “my turn” or “other’s turn.”
- Can MA2QL apply to non-cooperative MARL settings? (i.e., Markov Game)

Q2. Unclear description of ‘fair comparison’ between MA2QL and IQL

- In Multi-agent Mujoco, $K=8000$. In 2-agent HalfCheetah, for example, for $2K=16000$ environment steps, each agent is trained with MA2QL. Then how IQL is fairly trained? Are the two agents simultaneously trained every 2 environment steps during $2K=16000$ environment steps?
- I am also confused with MPE and SMAC. At agent $i$'s turn, each agent $j$ stores the $m$ transition samples ($m$ environment steps), and agent $i$ conducts $K$ training steps. In total, each of $n$ agents conducts $K$ training steps for $mn$ environment steps.
Then how IQL is fairly trained? I think IQL agents simultaneously update Q-function one step (i.e., one gradient step) every $\frac{mn}{K}$ environmental step so that during $mn$ environment steps, each of $n$ agents conducts $K$ training steps.
If my understanding is correct, what is the value of $m$ in MPE and SMAC? What is the effect of the value of $m$? Is IQL properly tuned in this setting?
 - Clarification with pictorial examples can help to understand.


Q3. Regarding sampling issues in learning Q-network

As stated on page 5, when learning Q-network, MA2QL permits containing samples from past transitions in a single replay memory. The authors did not provide any remedy reducing the gap between theoretical on-policy learning and single replay memory usage. It would be better to try any remedy to reduce the logical gap.

Q4. Multiple Nash equilibria

The main claim relies on the assumption that there is a unique Nash equilibrium, and the authors described a short remedy dealing with multiple Nash equilibria on page 4. Can we verify the remedy experimentally? For example, we may construct a matrix game with multiple Nash equilibria to check whether the proposed method can be extended in practical cases.


Q5. Comparison with hysteretic Q-learning

The authors need to compare MA2QL with hysteretic Q-learning since hysteretic Q-learning is pre-existing baseline in fully decentralized learning. The considered environments by **Zhang et al. (2020) are different from those in this paper.
** Zhang et al., Bi-Level Actor-Critic for Multi-Agent Coordination, AAAI 20.


Q6. Environments
- Can MA2QL be effective with Multi-agent Mujoco when agent_obsk=0?
- Please explicitly write whether each environment is partially observable or not in each title in Figure 4 for readers.
- In SMAC, is there any case where IQL fails but only MA2QL succeeds?

Q7. Reproducibility

The authors did not provide their implementation codes for verification.


Notation: $\gamma$ is missing in the third paragraph of Section 5.2.


**Summary Of The Paper:**

In this paper, the authors consider a fully decentralized cooperative multi-agent reinforcement learning (MARL). To alleviate the non-stationarity in decentralized MARL, this paper proposes to learn agents by letting each agent take turns updating each action-value function. Based on this simple idea, multi-agent alternate Q-learning (MA2QL) is proposed for practical implementation, and a theoretical explanation is described. Numerical results show that MA2QL can perform better than independent Q-learning in the considered cooperative MARL environments under fully decentralized settings.

**Summary Of The Review:**

It is crucial to study new learning methods in fully decentralized multi-agent reinforcement learning. However, the technical novelty of the newly developed algorithm remains questionable, and there are many unclear aspects of the core idea of alternate learning. Discussions and experiments should be expanded. Thus, a major revision is necessary both theoretically and experimentally.

---

> ### Author Response · Authors · 2022-11-15
> **Response**
>
> > about alternate learning
>
> Learning on an extremely large distributed system is hard for IQL, MA2QL, and CTDE methods. For such a system, one should consider methods like on mean-field or networked MARL.
>
> The update of IQL does not guarantee the improvement of the joint policy. **So, you cannot simply say more gradient steps will lead to better performance. Otherwise, there is no non-stationary problem at all.**
>
> Theorem 1 holds for any order at each round, which means the order does not affect the convergence *theoretically*. In experiments, we use a randomly determined order throughout a training process.  In practice, it is convenient to use a pre-defined order, which is indeed mentioned in Section 6. We performed additional experiments on the didactic game using different orders. For the two pre-defined orders, their performance is almost the same. For random order at each round, MA2QL still outperforms IQL. The results are included in Appendix F. Note that random order requires all agents to reach an agreement on who should update at each round. thus it is much less practical.
>
> MA2QL *cannot* apply to Markov game.
>
>
> >  **whether the comparison is fair**
>
> Yes, the comparison is fair. Note that **IQL is the anchor**, not MA2QL.  We chose the well-tuned number of gradient steps per environment step/episode for IQL,  then adjusted the setting for MA2QL accordingly to make sure the gradient steps of each MA2QL agent are the same as IQL agent.
>
> For MPE and SMAC, agents learn on each episode, not on each environment step, since the network is based on RNN, as mentioned in Appendix C.2. For these two environments, $K$ is the number of episodes.
>
> Finally, to make sure you understand this correctly, we give an example of SMAC. Suppose there are $n$ agents and
> each IQL agent updates its Q-network by $m$ gradient steps using mini-batch sampled from the replay buffer after interacting in the environment for an episode, where $m$ is well-tuned for IQL. Then, for a fair comparison, how do MA2QL agents learn?  We set $K=50$ for MA2QL in SMAC. Suppose it is the turn of agent $i$ to update. After each episode, agent $i$ updates its Q-network by $m\times n$ gradient steps. So, during its turn, it performs $m\times n \times K$ gradient steps.
>
> So, for $n \times K$ episodes. Each IQL agent has $m\times n \times K$ gradient steps, and each MA2QL also has $m\times n \times K$ gradient steps.
>
>
> > sampling issues in learning Q-network
>
> A simple solution is to control the size of replay buffer to remedy this. However, this is not the focus of this paper. Moreover, to bridge this gap, a more rigorous study is required, which is beyond the scope of this paper.
>
> > multiple Nash equilibria
>
> This is a misunderstanding. We do not assume a unique Nash equilibrium for the joint policy. We assume for each agent $i$, $Q_i^*(s,\cdot)$ has the unique best action. Our remedy to this is to add a positive random noise to the reward function. For a matrix game, adding noise can apparently differentiate actions in terms of reward. There is no need to check this out in a matrix game.
>
> > hysteretic Q-learning
>
> Hysteretic Q-learning performs equivalently as IQL in multi-agent MuJoCo and even worse in SMAC as shown in the concurrent work [1]. Thus, Zhang et al., 2020 and [1] reach the same conclusion that hysteretic Q-learning may not work well in practice. Besides, the core of MA2QL is alternate learning as you mentioned, such a principle can be built on other Q-learning variants, such as hysteretic Q-learning or [1]. Thus,  comparing with hysteretic Q-learning is orthogonal to the main contribution of MA2QL.
>
> [1] Best Possible Q-Learning, ICLR 2023 submission.
>
> > environments
>
> We have added experiments in multi-agent MuJoCo with agent_obsk=0. MA2QL still outperforms IQL. The learning curves are included in Appendix F.
>
> In SMAC, we found that if the training is long enough, e.g., 10M steps. IQL will not fail.
>
>
> > reproducibility
>
> We will release the code upon acceptance, but we guarantee the results are correct and the comparison is fair.

---

> > ### Author Response · Authors · 2022-11-18
> > **Follow-up**
> >
> > As the end of the rebuttal period is approaching, please let us know whether our responses have addressed your concerns or if you have additional comments.
> >
> > Thanks.

---

> > > ### Comment · Reviewer_J8oo · 2022-11-18
> > > **Thanks**
> > >
> > > I carefully read the reply regarding my questions. I will also discuss the main issues with the other reviewers. Thanks for the author's response.

---

### Decision · Program_Chairs · 2023-01-20

**Decision:**

Reject

**Justification For Why Not Higher Score:**

1) Novelty of the approach is highly limited since similar methods are well established.
2) Experimental evaluation is lacking - e.g. common baselines such as MAPPO are missing in SMAC and no statistical tests for significance are present.
3) It's also unclear that this issue is even a problem in e.g. MAPPO since parameter updates are already small.

**Justification For Why Not Lower Score:**

N/A

**Metareview: Summary, Strengths And Weaknesses:**

Summary:
The authors present a simple method for learning in Dec-POMDPs which does alternating updates for players rather than simultaneous updates for all players.
This is a simple fix for the "non-stationarity" issue in MARL. This issue is particularly relevant for Q-learning based methods, since the epsilon greedy action selection means small parameter update can lead to large policy changes.

Strength:
Very simple method with solid theory and simple implementation.

Weakness:
Novelty of the approach is highly limited since similar methods are well established.
Experimental evaluation is lacking - e.g. common baselines such as MAPPO are missing in SMAC and no statistical tests for significance are present.
It's also unclear that this issue is even a problem in e.g. MAPPO since parameter updates are already small.

I will also say that the authors clarified a lot of the concerns of the reviewers and that incorporating these clarifications into the draft will likely improve the chance of success of the paper at a future conference.